# Relationship between adiponectin multimer levels and subtypes of cerebral infarction

Noriko Tagawa[1]☯, Aya Fujinami[2]☯, Shigeatsu Natsume[3‡], Shigeto Mizuno[4‡], Ikuo Kato[1]☯ *

1 Laboratory of Medical Biochemistry, Kobe Pharmaceutical University, Kobe, Japan, 2 Comprehensive Education and Research Center, Kobe Pharmaceutical University, Kobe, Japan, 3 Rehabilitation Department, Eishokai, Yoshida Hospital, Kobe, Japan, 4 Endoscopy Department, Kindai University Nara Hospital, Ikoma, Japan

☯ These authors contributed equally to this work.
‡ SN and SM also contributed equally to this work.
* i-kato@kobepharma-u.ac.jp

## Abstract

### Aim

Serum adiponectin levels are decreased in patients with cerebral infarction. Adiponectin in circulation exists in three isoforms: high molecular weight (HMW), medium molecular weight (MMW), and low molecular weight (LMW) adiponectin. We measured serum levels of total adiponectin and adiponectin multimers (HMW, MMW, and LMW) in patients with cerebral infarction and compared the serum levels of the three adiponectin multimers in stroke subtypes. We also evaluated the clinical value of adiponectin multimer levels as a biomarker for cerebral infarction.

### Methods

We assessed a total of 132 patients with cerebral infarctions. The serum levels of total and adiponectin multimers were measured using enzyme-linked immunosorbent assay (ELISA).

### Results

The total and HMW adiponectin levels were significantly lower in atherothrombotic infarction (AI) than in cerebral embolism (CE) (total, $p < 0.05$; HMW, $p < 0.05$). In male patients, the MMW adiponectin level was significantly lower in the lacunar infarction (LI) group than in the AI group ($p < 0.05$). The LMW adiponectin level was significantly lower in the AI group than in the LI and CE groups (LI, $p < 0.001$; CE, $p = 0.001$). However, there were no significant differences in adiponectin multimer levels among the stroke subtypes in female subjects. Additionally, in female patients with AI and LI, the LMW adiponectin levels were negatively associated with C-reactive protein (CRP; AI, $p < 0.05$; LI, $p < 0.05$).

**Data Availability Statement:** All relevant data are within the manuscript. We make all data underlying the findings in this manuscript fully available. We present those data as S1 Table.

**Funding:** Shigeto Mizuno (SM) received Grants-in-Aid for Scientific Research (25460233) from the Ministry of Education, Culture, Sports, and Technology of Japan (https://www.mext.go.jp/english/). The funders had no role in study design, data collection and analysis, decision to publish, or preparation of the manuscript. "IK (Ikuo Kato) received Center for the Advanced Research and Technology of Kobe Pharmaceutical University (award number: none) from the Association of Private Universities of Japan. The funders had no role in study design, data collection and analysis, decision to publish, or preparation of the manuscript".

**Competing interests:** The authors have declared that no competing interests exist.

## Conclusion

These findings suggest that a decrease in adiponectin is associated with AI and that serum LMW adiponectin level represents a potential biomarker for AI.

## Introduction

Cerebral infarction is associated with metabolic syndrome and often caused by atherosclerosis [1, 2]. Metabolic syndrome represents a cluster of independent risk factors for cardiovascular diseases. The major factors include high blood pressure, dyslipidemia, and glucose metabolism disorder, and is associated with a high risk of cerebral infarction [3].

Adiponectin is an adipokine with anti-atherogenic [4, 5], anti-inflammatory [6] and anti-diabetic properties and takes part in the regulation of several metabolic processes. Serum adiponectin levels are lower in obese subjects compared to non-obese subjects [7] and negatively correlate with body mass index (BMI) [8]. Numerous studies have shown that adiponectin produces anti-inflammatory effects [9] and inhibits TNF-alpha secretion [10]. Hence, adiponectin plays an important role in suppressing the onset and development of cardiovascular diseases.

Adiponectin circulates in the plasma as low molecular weight (LMW), medium molecular weight (MMW), and high molecular weight (HMW) forms [11]. Studies have shown that HMW adiponectin suppresses the onset of cardiovascular diseases, decreases weight [12], and improves insulin resistance [13]. HMW adiponectin has been suggested to be the most active form because it activates adenosine 5'-monophosphate (AMP)-activated protein kinase and has anti-apoptotic activity in vascular endothelial cells [14]. Several studies have focused on total- and HMW adiponectin levels, and the HMW-to-total adiponectin ratio in type 2 diabetes, cardiovascular diseases, and cerebral infarction. However, the roles of the multimeric forms of adiponectin in these diseases remains controversial [15–18]. Baranowska *et al.* reported that serum levels of adiponectin and adiponectin multimer were significantly reduced in women with acute ischemic stroke relative to those in matched controls [19].

Plasma adiponectin levels have been shown to be lower in patients with cerebral infarction compared to that in control subjects [20]. However, studies on the relationships between the adiponectin forms (total, HMW, MMW, and LMW) and the clinical subtypes of cerebral infarction are limited.

Therefore, the aim of this study was to investigate the association between the subtypes of cerebral infarction and adiponectin forms (total, HMW, MMW, and LMW), and to assess the clinical value of adiponectin multimer levels as biomarkers for cerebral infarction.

## Subjects and methods

### Subjects

A total of 132 patients with cerebral infarction who were enrolled between November 2008 and April 2009 were included in the study. All patients were hospitalized at Eishokai Yoshida Hospital (Kobe, Japan) because of the onset of acute stroke. The subjects included 50 patients with atherothrombotic infarction (AI), 45 patients with lacunar infarction (LI), and 37 patients with cerebral embolism (CE) (Table 1). Clinical characterization of the cerebral infarction was performed using magnetic resonance imaging including T2-weighted imaging (T2WI) and diffusion tensor imaging (DTI). The study protocol was approved by the Ethics Committee of

**Table 1. Physical and metabolic characteristics of the subjects.**

|  | AI | LI | CE | p value | | |
|---|---|---|---|---|---|---|
|  |  |  |  | AI vs. LI | AI vs. CE | LI vs. CE |
| Subjects (n) | 50 | 45 | 37 |  |  |  |
| Sex, male/female | 31/19 | 28/17 | 20/17 |  |  |  |
| Age (years) | 74.0±11.2 | 73.9±10.3 | 75.9±10.6 |  |  |  |
| Day from onset (day) | 2.51±4.72 | 2.73±4.32 | 2.97±5.83 |  |  |  |
| Body mass index (kg/m$^2$) | 23.4±4.8 | 23.2±3.2 | 22.0±3.1 |  |  |  |
| Systolic blood pressure (mmHg) | 157±25 | 162±19 | 155±27 |  |  |  |
| Diastolic blood pressure (mmHg) | 85±16 | 85±14 | 88±16 |  |  |  |
| Total cholesterol (mg/dL) | 210±49 | 208±44 | 203±44 |  |  |  |
| LDL cholesterol (mg/dL) | 127±41 | 125±37 | 120±40 |  |  |  |
| HDL cholesterol (mg/dL) | 58±17 | 56±17 | 66±20 |  |  |  |
| Triglycerides (mg/dL) | 131±74 | 135±79 | 88±35 |  | 0.0177 | 0.0102 |
| High sensitive CRP (mg/dL) | 0.16±0.19 | 0.14±0.21 | 0.22±0.24 |  |  |  |

AI, atherothrombotic infarction; LI, lacunar infarction; CE, cerebral embolism; LDL, low-density lipoprotein; HDL, high-density lipoprotein; CRP, C-reactive protein.
Data are presented as mean ± standard deviation (SD).

our institution, and the study was conducted in accordance with the ethical principles of the Helsinki Declaration. Written informed consent was obtained from all the study participants. All serum samples from those 132 subjects were stored since November 2008. Adiponectin assays were performed during May and Jun in 2013.

## Clinical and laboratory assessments

Serum samples were collected from the patients after admission to the hospital. Venipuncture blood was directly collected into a tube containing aprotinin, polypeptide protease inhibitor and was immediately centrifuged at 4 ˚C. Serum sample was divided into tubes in a 0.1-mL aliquot and stored at -80 ˚C until used. The serum levels of total and multimeric adiponectin were measured in duplicate using an enzyme-linked immunosorbent assay (ELISA) kit (SEKISUI MEDICAL, Tokyo, Japan) according to the manufacturer's instructions. Briefly, the adiponectin levels of total, HMW and (HMW and MMW) were measured by this ELISA kit. MMW adiponectin levels were calculated by subtracting HMW from (HMW and MMW). LMW levels were calculated by subtracting (HMW and MMW) from total adiponectin. The intra- and inter-assay coefficients of variation for multimeric adiponectin ELISA kits were < 5.4% and < 6.0%, respectively. The levels of total cholesterol (T-Cho), low-density lipoprotein cholesterol (LDL-Cho), high-density lipoprotein cholesterol (HDL-Cho), triglyceride (TG), and C-reactive protein (CRP) were measured. BMI was calculated based on the patient's height and weight as follows: BMI = weight/height$^2$ [kg/m$^2$]. The modified Rankin scale (mRS) scores were assessed as prognostic indices of stroke (no symptoms, 0; severe disability, 5; death, 6).

## Statistical analysis

Data are shown as mean ± standard deviation (SD). The serum levels of adiponectin were not normally distributed, as shown by the Kolmogorov-Smirnov test. Therefore, the statistical comparison of variables between the groups was performed using the Kruskal-Wallis test and the Mann-Whitney U test with Bonferroni correction. Additionally, the degree of correlation

between the selected variables was determined using Spearman's correlation analysis. Statistical significance was set at $P < 0.05$. Statistical analyses were performed using Statview version 5.0 (Abacus Concepts, Berkeley, CA, USA).

## Results

### Characteristics of the patients

The patient characteristics are shown in Table 1 (Individual data including serum adiponectin levels were presented in S1 Table). Serum TG levels were higher in patients with AI and LI compared to that in patients with CE. Significant differences were observed in the TG levels: AI vs. CE ($p = 0.0177$) and LI vs. CE ($p = 0.0102$). There were no significant differences in any other parameter between patients with AI, LI, and CE.

### Relationship between adiponectin forms and stroke subtypes

Fig 1 shows the serum adiponectin level in patients with cerebral infarction. The total and HMW adiponectin levels were significantly lower in AI than in CE (7.1 ± 3.8 vs. 9.7 ± 5.3 μg/mL, respectively; $p = 0.0312$ and 3.5 ± 2.4 vs. 5.2 ± 3.5 μg/mL, respectively; $p = 0.0444$). The serum LMW adiponectin level was lower in AI than that in LI and CE (2.1 ± 0.9 vs. 3.0 ± 1.2 μg/mL, respectively; $p = 0.0003$ and 2.1 ± 0.9 vs. 3.0 ± 1.1 μg/mL, respectively; $p = 0.001$), whereas no significant differences in the MMW adiponectin levels were found among the stroke subtypes.

Furthermore, we compared the levels of adiponectin multimer forms between male and female patients with cerebral infarction. Differences in levels between the sexes were noted except for LMW adiponectin levels in patients with LI and CE (S2 Table). In male patients, MMW adiponectin levels were significantly lower in the LI group than in the AI group (0.6 ± 0.5 vs. 1.1 ± 0.6 μg/mL, respectively; $p = 0.0366$).

The LMW adiponectin level in AI was significantly lower than that in LI and CE (1.7 ± 0.8 vs. 2.9 ± 1.3 μg/mL, respectively; $p = 0.003$ and 1.7 ± 0.8 vs. 3.0 ± 1.2 μg/mL, respectively; $p = 0.0003$) (Fig 2). However, there were no significant differences in the total, HMW, MMW, and LMW adiponectin levels in female patients (Fig 3).

### Correlation between the adiponectin forms and clinical parameters

Table 2 shows the correlation between total (Table 2a), HMW (Table 2b), MMW (Table 2c), and LMW (Table 2d) adiponectin levels and various characteristics of patients with AI, LI, and CE.

Total adiponectin levels were positively correlated with age and HDL-Cho in patients with AI and LI ($p = 0.0058$, and $p = 0.0014$, respectively; $p = 0.0005$ and $p = 0.0441$, respectively), with mRS score in the patients with AI ($p = 0.0391$), and with age in the LI male patients ($p = 0.0009$). Total adiponectin levels showed significant negative correlation with BMI in patients with AI, LI, and CE ($p = 0.0274$, $p = 0.0077$, and $p = 0.0398$, respectively) and in LI male and female patients ($p = 0.0060$, and $p = 0.0282$, respectively), with diastolic blood pressure (dBP) in the LI all and male patients ($p = 0.0115$ and $p = 0.0017$, respectively), with TG in the AI all and male patients ($p = 0.0397$ and $p = 0.0457$, respectively), systolic blood pressure (sBP) in the patients with LI ($p = 0.0420$), and with T-cho and LDL-cho in male patients with CE ($p = 0.0155$, $p = 0.0205$, respectively).

HMW adiponectin levels was significantly positively correlated with age in the patients with AI and LI and the LI male patients ($p = 0.0124$, $p < 0.0001$, and $p = 0.0004$, respectively), with HDL-Cho in the patients with AI and CE ($p = 0.0072$, and $p = 0.0219$, respectively), and

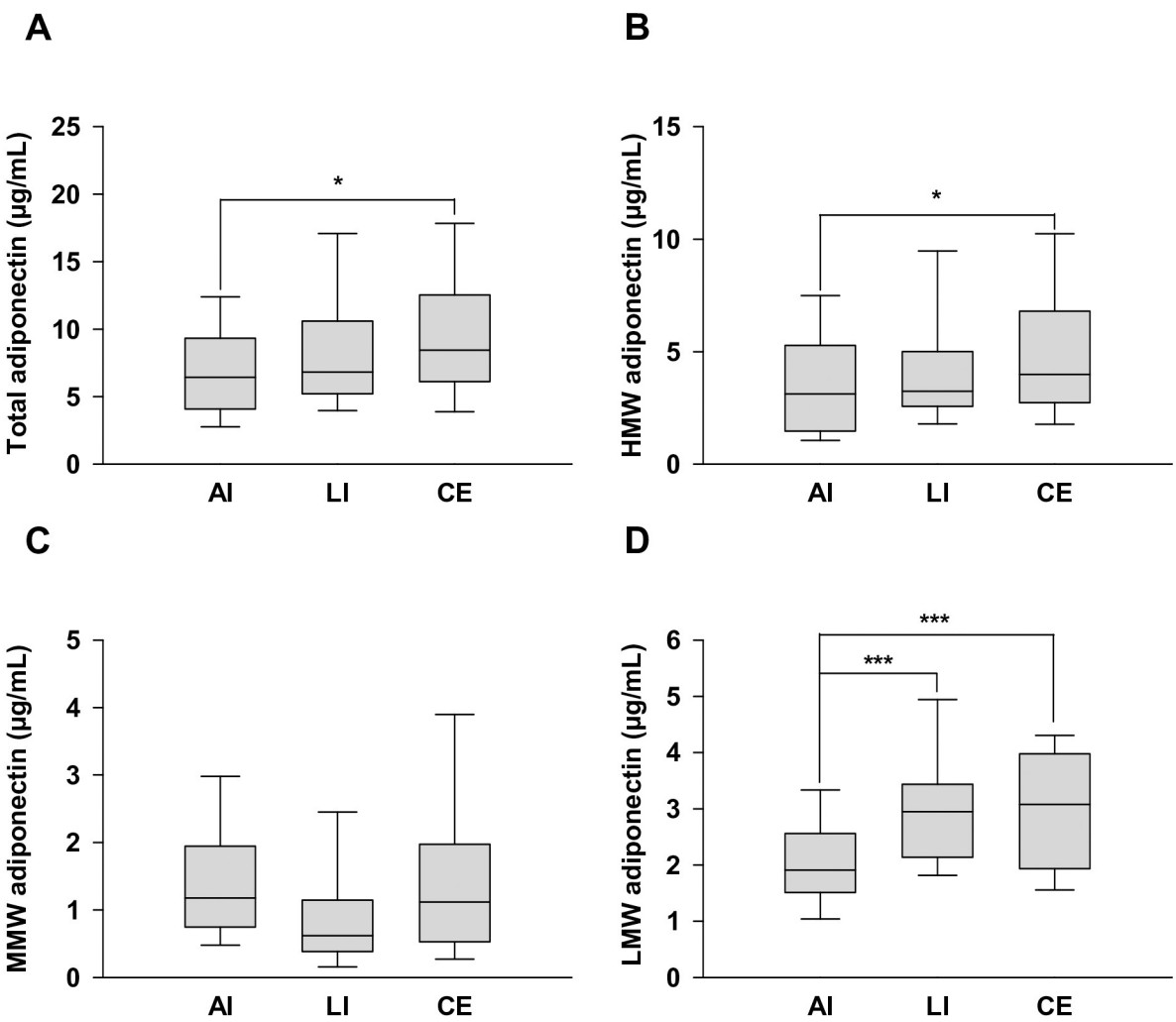

**Fig 1. Serum adiponectin multimer levels in patients with AI, LI, and CE.** Levels of (a) total adiponectin, (b) high molecular weight (HMW) adiponectin, (c) medium molecular weight (MMW) adiponectin, and (d) low molecular weight (LMW) adiponectin. Data are presented as the means ± standard deviation (SD). * $p < 0.05$, *** $p < 0.001$.

with mRS scores in the LI all and male patients ($p = 0.0186$, and $p = 0.0175$, respectively). HMW adiponectin levels showed significant negative correlation with BMI in patients with AI, LI, and CE ($p = 0.0380$, $p = 0.0046$, and $p = 0.0447$, respectively) and the LI male and female patients ($p = 0.0054$, and $p = 0.0211$, respectively), with TG in the patients with LI and the LI male and the CE male patients ($p = 0.0274$, $p = 0.0335$, and $p = 0.0334$, respectively), and dBP in the LI and male patients ($p = 0.0192$, and $p = 0.0030$, respectively).

MMW adiponectin levels showed significant positive correlation with age in the patients with AI and CE and the AI male patients ($p = 0.0278$, $p = 0.0135$, and $p = 0.0403$, respectively), with HDL-Cho in the patients with AI and LI ($p = 0.0439$, and $p = 0.0354$, respectively). MMW adiponectin levels had significant negative correlation with dBP in LI all and LI male patients ($p = 0.0202$, and $p = 0.0069$, respectively), with BMI in the patients with CE and the LI female patients ($p = 0.0330$ and $p = 0.0433$, respectively), sBP in patients with LI ($p = 0.0477$), and with T-cho in the CE male patients ($p = 0.0195$).

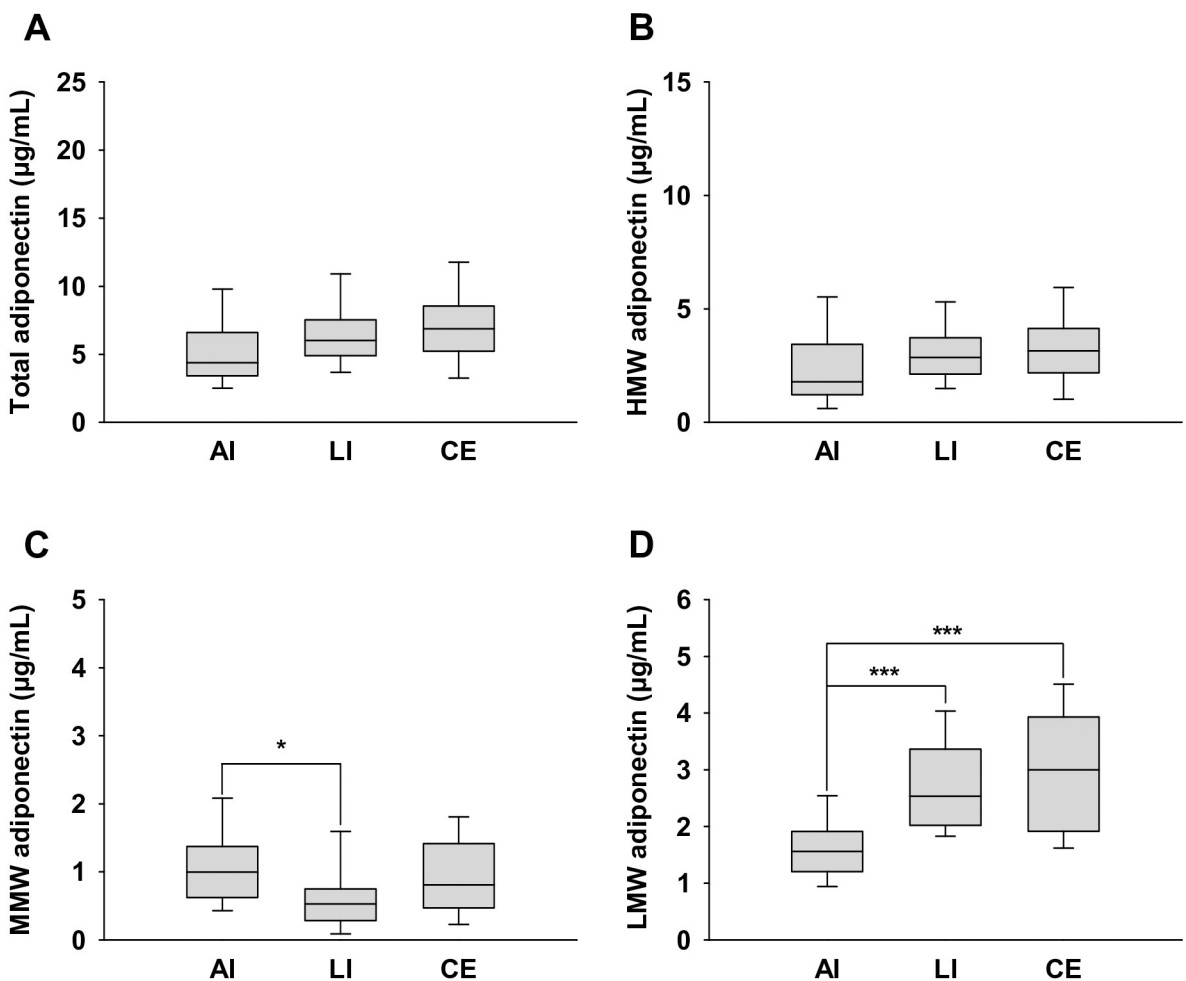

**Fig 2. Serum adiponectin multimer levels in male patients with AI, LI, and CE.** Levels of (a) total adiponectin, (b) high molecular weight (HMW) adiponectin, (c) medium molecular weight (MMW) adiponectin, and (d) low molecular weight (LMW) adiponectin. Data are presented as the means ± SD. * $p < 0.05$, *** $p < 0.001$.

LMW adiponectin levels had significant positive correlation with HDL-Cho in patients with AI and LI and AI male patients ($p < 0.0001$, $p = 0.0499$, and $p = 0.0083$, respectively), with age in the patients with AI and LI male patients ($p = 0.0026$, and $p = 0.0075$, respectively), and with mRS scores in the patients with AI ($p = 0.0103$). LMW adiponectin levels had significant negative correlation with hs-CRP in the AI female and the LI female patients ($p = 0.0432$ and $p = 0.0341$, respectively), with BMI in patients with AI and the AI male and the LI male patients ($p = 0.0024$, $p = 0.0264$, and $p = 0.0478$, respectively), with T-cho and LDL-cho in the CE male patients ($p = 0.0159$ and $p = 0.0150$, respectively), with dBP in the LI male patients ($p = 0.0071$), and with the mRS score in the LI female patients ($p = 0.0225$), and with TG in the AI all and male patients ($p = 0.0131$, and $p = 0.0200$, respectively).

## Discussion

In this study, we measured serum adiponectin multimer levels in subjects with cerebral infarction and compared the data to examine the relationship between the subtypes of cerebral infarction and serum levels of total, HMW, MMW, and LMW adiponectin. Furthermore, we

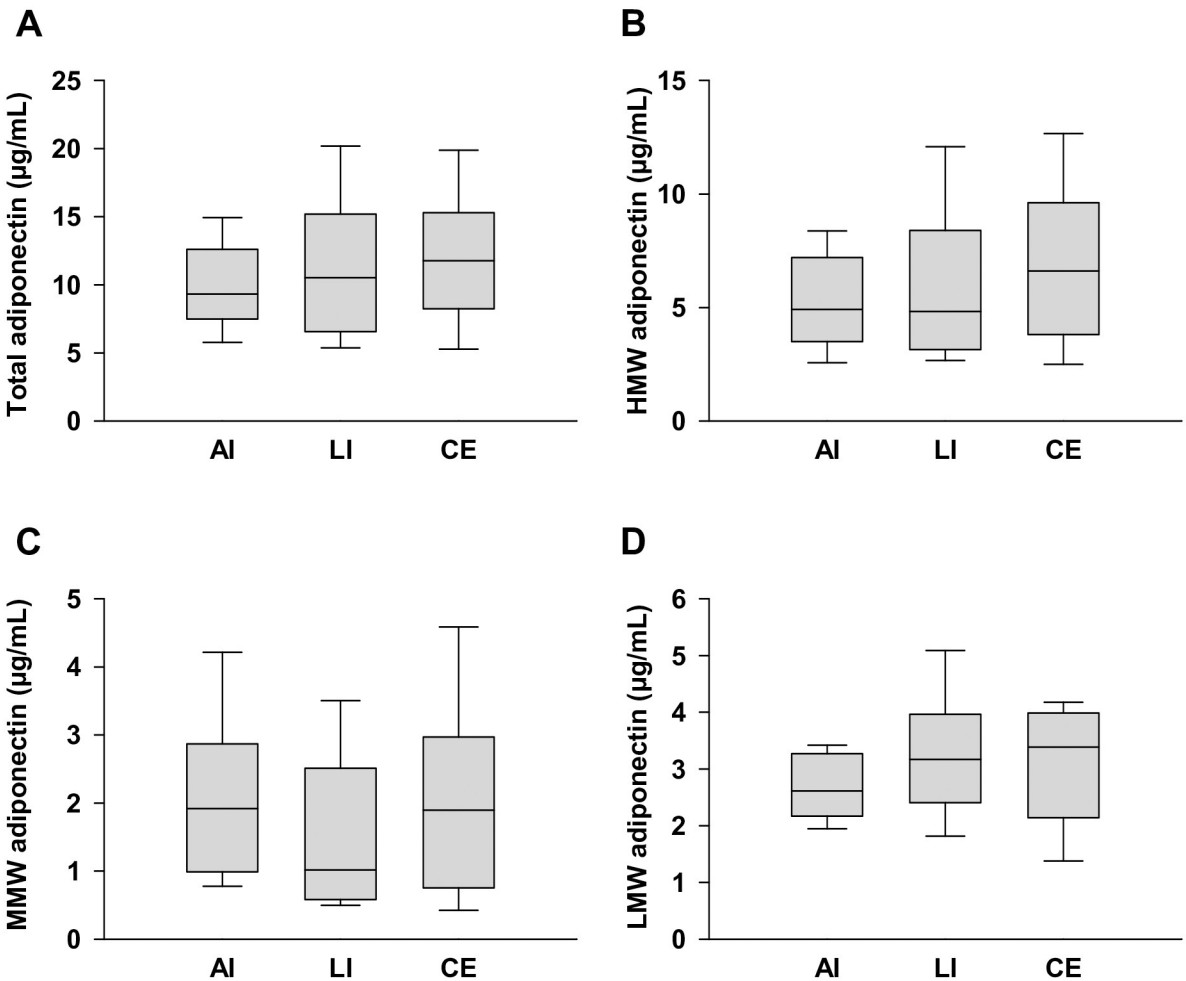

**Fig 3. Serum adiponectin multimer levels in female patients with AI, LI, and CE.** Levels of (a) total adiponectin, (b) high molecular weight (HMW) adiponectin, (c) medium molecular weight (MMW) adiponectin, and (d) low molecular weight (LMW) adiponectin. Data are presented as the means ± SD.

investigated the correlation between serum adiponectin multimer levels and clinical parameters.

We found that serum levels of total, HMW, and LMW adiponectin were lower in patients with AI than in those patients with LI and CE. In female patients, there were no significant differences in serum adiponectin multimer levels among the stroke subtypes. In view of our results, the reduction in serum total adiponectin levels in subjects with cerebral infarction may reflect a reduction in patients with AI. The serum level HMW adiponectin does not decrease in cerebral infarction [17]. This finding may reflect the results of serum HMW adiponectin levels in all patients. As noted in our results, HMW adiponectin levels may be lower in patients with AI compared to those with other subtypes when the measured values are compared among the stroke subtypes. To the best of our knowledge, the relationship between LMW adiponectin levels and cerebral infarction has not been investigated previously. MMW and LMW adiponectin have been found in human sera and cerebrospinal fluid (CSF) [21, 22]. Kusminski *et al.* showed that LMW adiponectin is the primary form of adiponectin in the human CSF [21]. Moreover, it has been shown that LMW adiponectin has anti-inflammatory activity that

**Table 2. The correlations of serum total, HMW, MMW, and LMW adiponectin levels with clinical parameters.**

| clinical parameter | AI | | | LI | | | CE | | |
|---|---|---|---|---|---|---|---|---|---|
| | all | male | female | all | male | female | all | male | female |
| (a) Total adiponectin levels | | | | | | | | | |
| Age (years) | 0.398 ** | 0.354 | −0.138 | 0.526 *** | 0.642 *** | 0.212 | 0.308 | −0.084 | 0.338 |
| BMI (kg/m²) | −0.329 * | −0.192 | −0.057 | −0.416 ** | −0.528 ** | −0.609 * | −0.348 * | −0.329 | −0.381 |
| T-Cho (mg/dL) | 0.212 | −0.212 | 0.363 | −0.056 | −0.117 | 0.010 | −0.247 | −0.587 * | 0.362 |
| LDL-Cho (mg/dL) | 0.085 | −0.183 | 0.158 | −0.152 | −0.212 | −0.106 | −0.313 | −0.562 * | 0.227 |
| HDL-Cho (mg/dL) | 0.471 ** | 0.154 | 0.086 | 0.310 * | 0.319 | 0.189 | 0.320 | 0.139 | 0.390 |
| TG (mg/dL) | −0.300 * | −0.365 * | 0.277 | −0.278 | −0.342 | −0.003 | −0.171 | −0.308 | −0.068 |
| sBP (mmHg) | 0.092 | −0.0004 | 0.116 | −0.307 * | −0.245 | −0.361 | 0.073 | −0.062 | 0.360 |
| dBP (mmHg) | −0.219 | −0.285 | 0.197 | −0.381 * | −0.605 ** | −0.129 | −0.059 | 0.098 | 0.177 |
| mRS score | 0.301 * | 0.178 | −0.278 | 0.250 | 0.352 | −0.087 | 0.269 | 0.160 | 0.206 |
| hs-CRP (mg/dL) | 0.129 | 0.236 | 0.145 | −0.223 | 0.080 | −0.435 | 0.037 | −0.068 | −0.094 |
| (b) HMW adiponectin levels | | | | | | | | | |
| Age (years) | 0.361* | 0.261 | −0.057 | 0.605 *** | 0.687 *** | 0.347 | 0.252 | −0.235 | 0.350 |
| BMI (kg/m²) | −0.309 * | −0.188 | −0.119 | −0.443 ** | −0.536 ** | −0.640 * | −0.339 * | −0.298 | −0.340 |
| T-Cho (mg/dL) | 0.221 | −0.185 | 0.439 | −0.139 | −0.206 | −0.110 | −0.178 | −0.448 | 0.374 |
| LDL-Cho (mg/dL) | 0.103 | −0.166 | 0.299 | −0.196 | −0.270 | −0.165 | −0.258 | −0.459 | 0.221 |
| HDL-Cho (mg/dL) | 0.396 ** | 0.053 | 0.071 | 0.276 | 0.276 | 0.128 | 0.412 * | 0.401 | 0.379 |
| TG (mg/dL) | −0.257 | −0.236 | 0.292 | −0.336 * | −0.409 * | −0.050 | −0.246 | −0.516 * | −0.068 |
| sBP (mmHg) | 0.074 | −0.036 | 0.103 | −0.273 | −0.204 | −0.354 | 0.051 | −0.084 | 0.293 |
| dBP (mmHg) | −0.232 | −0.279 | 0.150 | −0.353 * | −0.570 ** | −0.157 | −0.108 | 0.141 | 0.065 |
| mRS score | 0.267 | 0.175 | −0.256 | 0.363* | 0.475 * | 0.035 | 0.228 | 0.099 | 0.228 |
| hs-CRP (mg/dL) | 0.117 | 0.231 | 0.115 | −0.192 | 0.138 | −0.447 | 0.020 | −0.031 | −0.121 |
| (c) MMW adiponectin levels | | | | | | | | | |
| Age (years) | 0.318 * | 0.374 * | −0.219 | 0.227 | 0.058 | 0.006 | 0.418 * | 0.139 | 0.409 |
| BMI (kg/m²) | −0.094 | 0.040 | 0.110 | −0.206 | −0.172 | −0.560 * | −0.360 * | −0.106 | −0.484 |
| T-Cho (mg/dL) | 0.148 | −0.030 | 0.083 | −0.047 | −0.122 | 0.130 | −0.299 | −0.566 * | 0.109 |
| LDL-Cho (mg/dL) | 0.055 | 0.006 | −0.185 | −0.083 | −0.151 | 0.102 | −0.222 | −0.402 | 0.165 |
| HDL-Cho (mg/dL) | 0.297 * | 0.022 | 0.151 | 0.325 * | 0.199 | 0.356 | 0.114 | 0.009 | 0.116 |
| TG (mg/dL) | −0.159 | −0.140 | 0.125 | −0.288 | −0.240 | −0.244 | −0.192 | −0.252 | −0.032 |
| sBP (mmHg) | 0.253 | 0.103 | 0.391 | −0.298 * | −0.240 | −0.319 | 0.067 | −0.238 | 0.425 |
| dBP (mmHg) | −0.088 | −0.074 | 0.163 | −0.350 * | −0.520 ** | −0.191 | 0.027 | 0.034 | 0.425 |
| mRS score | 0.210 | 0.157 | −0.060 | 0.275 | 0.274 | 0.105 | 0.122 | 0.039 | 0.069 |
| hs-CRP (mg/dL) | 0.216 | 0.215 | 0.349 | −0.163 | 0.132 | −0.268 | −0.120 | −0.335 | −0.144 |
| (d) LMW adiponectin levels | | | | | | | | | |
| Age (years) | 0.435 ** | 0.318 | −0.017 | 0.283 | 0.515 ** | −0.101 | −0.066 | −0.109 | −0.256 |
| BMI (kg/m²) | −0.452 ** | −0.427 * | −0.090 | −0.301 | −0.381 * | −0.380 | −0.112 | −0.349 | 0.187 |
| T-Cho (mg/dL) | 0.218 | −0.063 | 0.216 | 0.120 | 0.045 | 0.256 | −0.155 | −0.585 * | 0.489 |
| LDL-Cho (mg/dL) | 0.099 | −0.105 | 0.207 | −0.033 | −0.069 | 0.041 | −0.222 | −0.621 * | 0.325 |
| HDL-Cho (mg/dL) | 0.589 *** | 0.490 ** | −0.111 | 0.303 * | 0.336 | 0.168 | 0.157 | −0.044 | 0.472 |
| TG (mg/dL) | −0.362 * | −0.425 * | 0.140 | −0.126 | −0.237 | 0.232 | −0.068 | −0.044 | −0.114 |
| sBP (mmHg) | −0.049 | −0.067 | −0.210 | −0.246 | −0.255 | −0.135 | 0.073 | −0.103 | 0.289 |
| dBP (mmHg) | −0.250 | −0.263 | 0.098 | −0.276 | −0.518 ** | 0.226 | 0.065 | 0.103 | −0.011 |
| mRS score | 0.374 * | 0.273 | −0.270 | −0.111 | 0.063 | −0.570 * | 0.279 | 0.272 | 0.262 |

(*Continued*)

**Table 2.** (Continued)

| clinical parameter | AI | | | LI | | | CE | | |
|---|---|---|---|---|---|---|---|---|---|
| | all | male | female | all | male | female | all | male | female |
| hs-CRP (mg/dL) | 0.016 | 0.294 | −0.477 * | −0.292 | −0.113 | −0.547 * | 0.096 | 0.114 | 0.033 |

Values are Spearman's correlation coefficient.

* $p < 0.05$,

** $p < 0.01$,

*** $p < 0.001$.

AI, atherothrombotic infarction; LI, lacunar infarction; CE, cerebral embolism; BMI, body mass index; T-Cho, total cholesterol; LDL, low-density lipoprotein; HDL, high-density lipoprotein; TG, triglyceride; sBP, systolic blood pressure; dBP, diastolic blood pressure; mRS, modified Rankin scale; HS, high sensitivity; CRP, C-reactive protein.

suppresses IL-6 secretion and induces IL-10 production [2]. Considering our finding that LMW adiponectin levels were lower in AI than in LI and CE, we speculate that the decrease in serum LMW adiponectin levels in AI patients may be due to translocation of LMW adiponectin for central adiponectin action. The change in adiponectin multimer levels, except for MMW adiponectin, may be a result of the development of AI.

To date, there have been few reports on the function of MMW adiponectin. We found that serum MMW adiponectin levels were lower in patients with LI than in those with AI and CE. In female patients, there were no significant differences in the serum levels of MMW adiponectin among the stroke subtypes. The decrease in serum MMW adiponectin levels in male patients with LI may help to distinguish LI from other stroke subtypes. Taken together, adiponectin multimers that are influenced by stroke subtypes may vary. Our findings that serum levels of total, HMW, and LMW adiponectin were lower in patients with AI than in those with LI and CE may reflect the levels in male patients. It is important to clarify this point in future studies.

In the present study, correlation analysis showed that serum adiponectin multimer levels were significantly correlated with age and HDL-Cho and negatively correlated with BMI. Human adiponectin levels have been shown to be lower in obese subjects than in non-obese subjects, and adiponectin is associated with lipid metabolism [23, 24]. In older subjects, circulating adiponectin levels increase with age [25, 26]. Therefore, our results were consistent with those of previous studies [25, 26]. High blood pressure was the most consistent risk factor for LI. Our finding of the negative correlation between serum total, HMW, and MMW adiponectin levels and sBP or dBP fit well with the cause of LI. The mRS score is considered a prognostic indicator of stroke. To preliminarily examine the relationship between adiponectin levels and disability or prognosis in cerebral infarction we chose mRS that has been most widely applied for evaluating recovery from stroke and is also commonly used as a functional prognosis evaluation item in clinical studies. However, mRS is based on the subjective evaluation of the evaluator and it is also known that there are considerable fluctuations. In the present study, for consistency of validity and reliability of the mRS, scoring with interview was performed by cerebral stroke specialists. The positive correlations among serum total, HMW, and LMW adiponectin levels and mRS score suggest that these adiponectins may be compensatorily upregulated in advanced disease stages.

The positive correlations among serum total, HMW, and LMW adiponectin levels and mRS score suggest that these adiponectins may be compensatorily upregulated in advanced disease stages. Therefore, it is difficult to directly determine the prognosis and dysfunction from the adiponectin levels, and it is necessary to consider various factors and interpret the relation.

This study examined serum levels of multimeric adiponectin at only one point after onset. Other studies have reported improvement of serum adiponectin levels and functional recovery by post-stroke treatment and rehabilitation. Biomarkers should be useful for disease detection, prognosis determination, follow-up, and therapeutic effect determination. It is necessary that comparison of serum multimeric adiponectin levels between after onset and after a period when functional recovery is observed [27–30].

In subjects with cerebral infarction, we also found a negative correlation between serum LMW adiponectin and hs-CRP levels. However, this relationship has been reported in female patients with AI and LI. Ridker *et al*. [31] demonstrated that the serum CRP level was slightly higher in myocardial infarction or symptomatic ischemic cerebrovascular disease before onset. The group with high serum hs-CRP levels had a high frequency of ischemic cerebrovascular disorders. hs-CRP is regarded as an independent predictor of symptomatic cerebral vascular disorders. Atherosclerosis has been attributed to damage to the vascular endothelium resulting from hypertension, hyperglycemia, oxidative LDL, inflammatory substances, free radicals, and smoking. As a result of the damage to endothelial cells, adhesion molecules, such as monocyte chemoattractant protein-1, intercellular adhesion molecule-1 (ICAM-1), and vascular cell adhesion molecule-1 (VCAM-1), are expressed on endothelial cells and induce the adhesion and infiltration of monocytes into the vascular intima. Monocytes that infiltrate the vascular intima differentiate into macrophages, and the formation of foam cells from macrophages is induced by oxidized LDL. Proinflammatory cytokines produced by foam cells within the plaque contribute to localized inflammation. Serum CRP level is a marker that reflects the extent of atherosclerosis and is a useful predictor of ischemic cerebrovascular disease and ischemic heart disease [32]. However, recent studies have noted the possibility that CRP is not only a marker of inflammation but also an accelerator of atherosclerosis [33, 34]. Furthermore, CRP is known to be deposited in the inflammatory regions. Bhakdi *et al*. [35] found that CRP was deposited in oxidized LDL, which plays an important role in the formation of atherosclerosis, and that CRP activated the expression of VCAM-1 and ICAM-1 in endothelial cells.

HMW adiponectin, which mainly activates AMP kinase, has anti-apoptotic activity in endothelial cells [14], and suppresses cell proliferation. Therefore, HMW adiponectin is thought to be the major active form of this protein. On the other hand, MMW and LMW adiponectin are thought to be related to central actions [22]. However, there are few reports of MMW and LMW adiponectin being involved in the suppression of cardiovascular diseases and improvement of insulin resistance.

Hs-CRP is considered a predictive factor for cardiovascular events. The results of this study indicate that LMW adiponectin, which is not considered to be a major active form, is a biomarker of cerebral infarction because LMW adiponectin was negatively correlated with serum hs-CRP. Hs-CRP is an excellent marker of general inflammatory response, but it does not appear to be a specific marker of cerebral infarction. Our data indicate that LMW adiponectin is more useful than hs-HRP as a marker of cerebral infarction. Analyzing the correlation between adiponectin and various factors is an effective tool for searching for biomarkers. If adiponectin impairs the functional molecular process, it may be directly reflected and predict the diagnosis, prognosis, and course of the disease. In addition, it must be considered that the receptor or signal transduction of adiponectin is impaired. It is also necessary to know that when the optimal concentration range of serum adiponectin or the factor to be examined is narrow, the correlation becomes positive or negative due to abnormally low or abnormally high values.

There are several limitations to this study. First, the sample size was small. Second, there were no significant differences in any of the serum adiponectin multimer levels among the stroke subtypes in female patients. This study targeted patients aged 74 ± 11 years. Women of this age are considered menopausal. Reduction in estrogen due to menopause has been

associated not only with menopausal disorder, osteoporosis, and atrophic vaginitis, but also with cardiovascular diseases, such as myocardial infarction and stroke. The incidence of dyslipidemia in men younger than 50 years is higher than that in women, but its incidence in women older than 50 years increases rapidly and then exceeds that in men. In women in the menstruating age group, estrogen is thought to have a positive effect on lipid metabolism and suppress the development of arteriosclerosis. Menopause is associated with changes in body composition including changes in deposition of subcutaneous fat to visceral fat, independent of changes caused by aging. Similarly, the change in body fat distribution is also related to the development of endocrine and metabolic disorders, independent of the decline in estrogen. Visceral fat measurements have been performed as direct methods using computer tomography and magnetic resonance imaging as direct methods, but there are still limitations due to the lack of sensitive tools for measuring body composition. Therefore, future studies that include women in the menstruating age group are needed to confirm the relationship between adiponectin multimers and subtypes of cerebral infarction.

## Conclusion

In conclusion, we demonstrated that serum levels of total, HMW, and LMW adiponectin were lower in patients with AI than in those with CE. In female patients with AI and LI, there were negative correlations between serum LMW adiponectin levels and hs-CRP levels. Our data suggest that the decrease in adiponectin may be associated with AI and that serum LMW adiponectin level represents a useful biomarker for cerebrovascular disorders.

## Supporting information

**S1 Table. The serum levels of total and multimeric adiponectin, and clinical parameters in patients with atherothrombotic infarction (AI), lacunar infarction (LI) and cerebral embolism (CE).**
(PDF)

**S2 Table. The serum levels of total and multimeric adiponectin in patients with cerebral infarction.**
(PDF)

## Acknowledgments

The authors wish to thank Yumi Tomofuji for her technical assistance.

## Author Contributions

**Conceptualization:** Shigeto Mizuno, Ikuo Kato.

**Data curation:** Aya Fujinami, Shigeatsu Natsume.

**Formal analysis:** Noriko Tagawa, Aya Fujinami.

**Funding acquisition:** Shigeto Mizuno, Ikuo Kato.

**Investigation:** Noriko Tagawa, Aya Fujinami.

**Methodology:** Shigeto Mizuno.

**Project administration:** Ikuo Kato.

**Resources:** Shigeatsu Natsume.

**Supervision:** Ikuo Kato.

**Visualization:** Noriko Tagawa, Aya Fujinami.

**Writing – original draft:** Noriko Tagawa, Aya Fujinami.

**Writing – review & editing:** Noriko Tagawa, Ikuo Kato.

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
