## [Decision Letter · Decision Letter 0]

2 Nov 2021

PONE-D-21-26718Relationship between adiponectin multimer levels and subtypes of cerebral infarctionPLOS ONE

Dear Dr. Ikuo Kato,

Thank you for submitting your manuscript to PLOS ONE. After careful consideration, we feel that it has merit but does not fully meet PLOS ONE’s publication criteria as it currently stands. Therefore, we invite you to submit a revised version of the manuscript that addresses the points raised during the review process.

We look forward to receiving your revised manuscript.

Kind regards,

Paolo Magni

Academic Editor

PLOS ONE

Journal Requirements:

Additional Editor Comments:

The ms. addresses an interesting issue. However, the paper contains several critical issues that need to be fully addressed prior to consideration of he paper for publication.

Reviewers' comments:

Reviewer's Responses to Questions

**Comments to the Author**

1. Is the manuscript technically sound, and do the data support the conclusions?

Reviewer #1: Partly

2. Has the statistical analysis been performed appropriately and rigorously? 

Reviewer #1: Yes

3. Have the authors made all data underlying the findings in their manuscript fully available?

Reviewer #1: No

4. Is the manuscript presented in an intelligible fashion and written in standard English?

Reviewer #1: Yes

5. Review Comments to the Author

Reviewer #1: It is an interesting study that aimed to describe the relationship between adiponectin isoforms and stroke subtypes. There is relevance in the theme because stroke molecular biomarkers is still a controversial field in the literature.

Introduction was well written, the problem is clear, however the authors could provide some hypotheses. It will improve the manuscript rational.

About methods, I´m a little worried about when volunteers were included in the study. Authors described " a total of 132 patients with cerebral infarction who were enrolled between November 2008 and April 2009 were included in the study". Blood samples were stored since 2009? How could you ensure the sample integrity? Please, clarify when molecular assays were performed. In addition, please describe if there was a blind evaluator for groups.

Authors must describe when the samples were collected: 1 hour, 8 hours, 24 hours after stroke… The results can be affected according to time after stroke.

About ELISA, were samples assessed in duplicate or triplicate?

Also describe any method used for stroke image diagnosis.

About results, I believe authors should provide the real values of p in the text (Line 120 -Relationship between adiponectin forms and stroke subtypes; Line 153 Correlation between the adiponectin forms and clinical parameters). For figures and tables OK.

Regarding to discussion, try not to repeat the introduction content.

Please, could you bring more information about biomarkers and functional recovery? I think it´s an important aspect for discussion. mRS score needs more attention and interpretation. Some correlations are positive and others negative, could it be related to some type of U effect or increase in adiponectin resistance because of loss of receptors?

The lack of sensitive tools for measuring body composition as a limitation.

6. PLOS authors have the option to publish the peer review history of their article (what does this mean?). If published, this will include your full peer review and any attached files.

Reviewer #1: **Yes: **Thiago Luiz de Russo

---

## [Author Response · Author response to Decision Letter 0]

17 Dec 2021

We make all data underlying the findings in this manuscript fully available. We present those data as supplemental table 1.

---

## [Decision Letter · Decision Letter 1]

28 Dec 2021

Relationship between adiponectin multimer levels and subtypes of cerebral infarction

PONE-D-21-26718R1

Dear Dr. Kato,

We’re pleased to inform you that your manuscript has been judged scientifically suitable for publication and will be formally accepted for publication once it meets all outstanding technical requirements.

Kind regards,

Paolo Magni

Academic Editor

PLOS ONE

Additional Editor Comments (optional):

Reviewers' comments:

Reviewer's Responses to Questions

**Comments to the Author**

1. If the authors have adequately addressed your comments raised in a previous round of review and you feel that this manuscript is now acceptable for publication, you may indicate that here to bypass the “Comments to the Author” section, enter your conflict of interest statement in the “Confidential to Editor” section, and submit your "Accept" recommendation.

Reviewer #1: All comments have been addressed

2. Is the manuscript technically sound, and do the data support the conclusions?

Reviewer #1: Yes

3. Has the statistical analysis been performed appropriately and rigorously? 

Reviewer #1: Yes

4. Have the authors made all data underlying the findings in their manuscript fully available?

Reviewer #1: Yes

5. Is the manuscript presented in an intelligible fashion and written in standard English?

Reviewer #1: Yes

6. Review Comments to the Author

Reviewer #1: The main points suggested were addressed by the authors. they included more information about functional recovery and biomarkers.

7. PLOS authors have the option to publish the peer review history of their article (what does this mean?). If published, this will include your full peer review and any attached files.

Reviewer #1: **Yes: **Thiago Luiz Russo

---

## [Editor Report · Acceptance letter]

3 Jan 2022

PONE-D-21-26718R1 

Relationship between adiponectin multimer levels and subtypes of cerebral infarction 

Dear Dr. Kato:

I'm pleased to inform you that your manuscript has been deemed suitable for publication in PLOS ONE. Congratulations! Your manuscript is now with our production department. 

Kind regards, 

on behalf of

Prof. Paolo Magni 

Academic Editor

PLOS ONE